# Allogeneic HLA Humoral Immunogenicity and the Prediction of Donor-Specific HLA Antibody Development

**DOI:** 10.3390/antib13030061

**Published:** 2024-07-24

**Authors:** Vadim Jucaud

**Affiliations:** Terasaki Institute for Biomedical Innovation, Los Angeles, CA 91367, USA; vjucaud@terasaki.org

**Keywords:** human leukocyte antigen, immunogenicity, allorecognition, de novo donor-specific antibodies, epitopes

## Abstract

The development of de novo donor-specific HLA antibodies (dnDSAs) following solid organ transplantation is considered a major risk factor for poor long-term allograft outcomes. The prediction of dnDSA development is a boon to transplant recipients, yet the assessment of allo-HLA immunogenicity remains imprecise. Despite the recent technological advances, a comprehensive evaluation of allo-HLA immunogenicity, which includes both B and T cell allorecognition, is still warranted. Recent studies have proposed using mismatched HLA epitopes (antibody and T cell) as a prognostic biomarker for humoral alloimmunity. However, the identification of immunogenic HLA mismatches has not progressed despite significant improvements in the identification of permissible mismatches. Certainly, the prediction of dnDSA development may benefit permissible HLA mismatched organ transplantations, personalized immunosuppression, and clinical trial design. However, characteristics that go beyond the listing of mismatched HLA antibody epitopes and T cell epitopes, such as the generation of HLA T cell epitope repertoires, recipient’s HLA class II phenotype, and immunosuppressive regiments, are required for the precise assessment of allo-HLA immunogenicity.

## 1. Introduction

The development of de novo donor-specific human leukocyte antigen (HLA) antibodies (dnDSAs) following solid organ transplantation is a major risk factor for poor long-term allograft outcomes [1,2,3,4,5]. In renal transplantation, the incidence rate of dnDSAs rarely exceeds 20% 1-year post-transplantation, and it can reach 25% 5 to 8 years post-transplantation [6,7,8]. Several risk factors have been associated with developing dnDSAs, including high HLA mismatches, under-immunosuppression, non-adherence to treatment, or allograft inflammation [6,9,10]. Clearly, not all organ transplant recipients have the same risk of developing dnDSAs, yet the risk prediction of dnDSA development before transplantation remains to be fully characterized. Although standard-of-care immunosuppression therapies contribute to the minimization of dnDSA development [11,12,13], other computational strategies rely on the precise evaluation, at the molecular level, of HLA mismatches to predict their dnDSA-inducing capacity, or immunogenicity, and identify permissible mismatches [14,15].

Counting HLA mismatches does not accurately evaluate the degree of incompatibility of a donor-recipient pair [16] since molecular differences between donors’ and recipients’ HLA antigens, rather than whole antigens, are recognized by the recipient’s humoral immune system. In addition, not all HLA antigens have equal dnDSA-inducing capacity [17]. The humoral response to HLA mismatches, or allogeneic HLAs (allo-HLAs), is extremely variable in that not all mismatched HLAs have an equal probability of inducing the development of dnDSAs [18]. Among HLAs, HLA class II (HLA-II) mismatches appear to be more immunogenic than HLA class I (HLA-I) mismatches [7,10,19]. In particular, HLA-DQ dnDSAs were shown to be the most common antibodies developed post-transplantation, to be resistant to treatment, and to have the highest association with poor graft outcomes such as antibody-mediated rejection, transplant glomerulopathy, and allograft failure [6,10,20,21,22,23,24]. Recognizing the complexity and variability of donor-specific humoral responses, several computational tools were developed, integrating large amounts of immunogenetic and molecular data, to assess the immunogenicity of allo-HLA and to identify permissible mismatches that do not induce dnDSAs [25,26]. These computational tools are constructed to reflect the humoral alloimmune response or indirect allorecognition pathway that is responsible for the development of dnDSAs [27,28,29,30]. This humoral immune response is dependent on the ability of B cells to capture and process allo-HLAs into antigenic peptides that are subsequently presented to follicular helper CD4+ T cells (T_fh_), located in secondary lymphoid organs, such as the spleen, tonsils, and lymph nodes. The cognate CD4+ T cell helper function is required to form germinal centers (GCs) where B cells differentiate into IgG dnDSA-secreting plasma cells and memory B cells [31,32].

GCs are specialized microenvironments within secondary lymphoid organs where mature B cells undergo proliferation, somatic hypermutation, and selection for high-affinity antibodies [31,32]. In GCs, the generation of dnDSAs is orchestrated through a complex interplay of cellular and molecular mechanisms. The process begins when allo-HLAs are captured and internalized by naive B cells. Upon recognizing allo-HLAs via their B cell receptors, B cells internalize, process, and present allo-HLA-derived peptides on their surface in the context of HLA class II molecules. This antigen presentation is crucial for the subsequent interaction with T_fh_, which provides essential signals for B cell activation and differentiation. B cells proliferate rapidly within the GC and undergo somatic hypermutation of their immunoglobulin genes, mediated by the enzyme activation-induced cytidine deaminase [33,34]. This process generates a diverse repertoire of B cells with varying affinities for the allo-HLAs. T_fh_ cells provide critical survival and differentiation signals through the engagement of CD40 ligand (CD40L) on T_fh_ cells with CD40 on B cells and the secretion of cytokines (i.e., IL-21 and IL-4) [35]. These interactions between T_fh_ and B cells promote the selection of high-affinity B cells and their differentiation into long-lived plasma cells, which secrete dnDSAs, or memory B cells, which provide a rapid and robust response upon re-exposure to allo-HLAs.

Since this humoral alloimmune response is a dynamic process orchestrated by the B and T cell compartments of the recipient’s immune system, allo-HLAs must exhibit two important immunologic domains to induce a robust response: B and T cell epitopes [36]. B cell epitopes reflect the antigenic determinant recognized by B cell or DSA, and T cell epitopes reflect the antigenic determinant required for the activation of CD4+ T cell helper function. Given the polymorphism of HLA molecules, a wide range of B and T cell epitopes have been characterized. Accordingly, allo-HLAs can be described as a unique set of mismatched B and T cell epitopes, which could be used as a strategy to refine HLA matching due to epitope sharing between HLA molecules [29]. Thus, efforts to assess allo-HLA immunogenicity concentrated on the association between B cell or T cell epitopes and the development of IgG dnDSAs. Recent concomitant analyses of B and T cell epitopes had positive predictive capabilities toward dnDSA production. The association between B and T cell epitopes was weaker in dnDSA-positive recipients than in dnDSA-negative ones [19]. Similarly, the development of dnDSAs could be predicted accurately, but mostly revealing the prediction of allo-HLAs that may not induce a dnDSA rather than the ones that may [37]. Nonetheless, it is paramount—from a clinical perspective—to know if certain allo-HLAs have a low risk of inducing a dnDSA.

With recent technological advances, allo-HLA immunogenicity is assessed using physiochemical parameters (mismatched amino acids, electrostatic charge difference, and hydrophobicity charge difference) and functional ones (mismatched B and T cell epitopes) [14,25]. These quantitative parameters reflect the molecular differences between donor and recipient HLA molecules [38]. Unequivocally, the level of differences is correlated with the presence or level of dnDSAs [39,40], although each parameter alone does not fully reveal allo-HLA immunogenicity. Indeed, the overlap in the distribution of the parameters’ differences between allo-HLAs that induce a dnDSA and those that do not suggest that many factors are involved in the immunogenicity of allo-HLAs [41]. In addition, allo-HLAs with a high number of differences may not induce dnDSAs, and others with a low number of differences induce dnDSAs. Although several studies have demonstrated the impact of these parameters on allograft outcomes, their concomitant evaluation and their impact on allo-HLA immunogenicity is scarce [9,25,41,42]. Therefore, the prediction of allo-HLA immunogenicity remains to be fully elucidated.

Allo-HLA sensitization, characterized by the production of IgG dnDSAs and immunological memory, is a complex process tightly regulated by the recipient’s immune system while being driven by the molecular differences exhibited by allo-HLAs. The goal of this review is to understand and clarify allo-HLA immunogenicity assessment—from both B cell and T cell perspectives—by discussing the kinetics of this biological process with respect to the current computational approaches that evaluate allo-HLA immunogenicity. Lastly, this review will discuss the limitations of assessing allo-HLA immunogenicity and future directions.

## 2. B Cell/Antibody Epitopes: The First Activating Signal

The humoral alloimmune response begins with transplant surgery, which can induce inflammation, injury, and subsequent activation of innate and adaptive immunity [43]. This initial inflammatory and shock response can trigger the release of allo-HLAs by the allograft tissues into the recipient’s circulation [1,2,44]. Soluble allo-HLAs can travel to the secondary lymphoid organs where the recipient’s naïve B cell, expressing allospecific B cell receptors (allo-BCRs), can recognize and bind their cognate soluble allo-HLAs [31,45]. The allo-BCR recognizes a molecular patch on the surface of the allo-HLA corresponding to a mismatched B cell epitope (i.e., HLA antibody epitope or eplet). The captured allo-HLA may induce BCR signaling and promote allo-HLA uptake for further processing and presentation to allo-HLA-specific follicular helper CD4+ T cells (allo-T_fh_). The recognition of allo-HLA by the allo-BCR constitutes the first event required to initiate the humoral alloimmune response [39], and without it, the recipient’s naïve B cells may not differentiate into dnDSA-IgG-secreting plasma cells. Indeed, immunogenic allo-HLAs carry at least one mismatched eplet [16,25,46]. Indeed, without mismatched eplets, the long-term dnDSA-free survival remained at 100% in a cohort of 664 renal transplant patients [46]. Hence, the mandatory signal provided by the recognition of allo-HLA by the allo-BCR emphasizes the importance of mismatched eplets on allo-HLA immunogenicity and the development of dnDSAs [47].

Eplets are three-dimensional structures exposed on the molecular surface of allo-HLAs (with an area of 700–900 Å) that are recognized by HLA antibodies [48]. The footprint of HLA antibodies, or paratope, recognizes an amino acid configuration composed of an immunogenic epitope, crucial amino acid (essential for antibody binding), and indifferent amino acids [49]. However, eplets are defined by the amino acid residues critical for antibody binding, which can be exposed or not on the molecular surface in a continuous or discontinuous sequence (Figure 1). Interestingly, in some cases, eplet analyses demonstrated that the non-exposed eplets had the potential to explain the reactivity pattern of a DSA in a patient [50]. HLA molecules exhibit a unique set of eplets, but each eplet is frequently shared between HLA molecules [48,51], with some epitopes being private (exclusively found on one HLA molecule) and others public (shared by two or more HLA molecules). Only the mismatched eplets, exhibited by allo-HLAs, may trigger an alloimmune response. To identify eplets, several computational approaches have been developed over the years. HLAmatchmaker was one of the pioneer approaches developed by R. Duquesnoy in 2001 [52]. HLAmatchmaker uses the amino acid sequences and the tri-dimensional structure of HLA molecules to identify all potential mismatched eplets (also referred to as amino acid triplets) from the high-resolution HLA typing of a donor-recipient pair [47]. HLAmatchmaker can identify two types of eplets, namely: (1) antibody-verified epitopes that are experimentally well characterized and correspond to the reactivity of an HLA antibody that was developed in the setting of allo-HLA immunization; and (2) other theoretical epitopes that correspond to computationally predicted eplets (presence of mismatched amino acids and accessible for antibody binding), but not experimentally validated or described in human allosera.

Since eplets are hypothetically generated 3D topographical structure configurations defined by polymorphic amino acid residues on allo-HLA molecules, extensive efforts have been conducted to identify, characterize, and prove that eplets are recognized and bound by DSA [53,54]. Pioneer work in this area has been conducted by El-Awar et al. since 2006, where they have defined and reported the so-called “Terasaki epitopes” consisting of 194 HLA-I epitopes, including 56 cryptic epitopes, 83 HLA class II epitopes, and 7 MICA epitopes [48]. Today, the HLA Eplet Registry (www.epregistry.com.br (accessed on 2 April 2024)), created in 2012 [55], serves as the international database indexing theoretical and confirmed HLA eplets that are used to develop computational tools to determine the immunogenic potential of allo-HLAs. For example, web-based tools called, such as Epvix [56] or HLA-EMMA [57], were developed to simultaneously perform HLA-I and -II amino acid sequences comparison between donor and recipients in an automated fashion using the current eplet databases to determine polymorphic and accessible amino acid mismatches that are likely to be the target of BCR and DSA.

The quantitative measure of mismatched eplets allowed for the assessment of allo-HLA immunogenicity. Several studies demonstrated the association between the quantity of mismatched eplets and the development of dnDSAs, T cell and antibody-mediated rejection, chronic glomerulopathy, and all-cause graft loss [25,46]. For example, HLA-DR/DQ eplet mismatch load improved the correlation with dnDSA development compared to whole antigen mismatch. This approach allowed kidney allograft recipients to be stratified into low, intermediate, and high alloimmune risk categories. Nonetheless, allo-HLA immunogenicity using eplets relies solely on the antigenicity of allo-HLAs, thus not revealing accurate immunogenicity despite recent findings demonstrating the use of a HLA-DR/DQ mismatch as a potential prognostic biomarker for primary alloimmunity [46]. In that regard, the Cambridge HLA immunogenicity algorithm was developed to characterize allo-HLAs by their difference in physiochemical properties, namely their mismatched amino acid, electrostatic charge, and hydrophobicity differences [40,58,59]. This algorithm relies on the fact that the uptake of allo-HLAs by B cells depends on the BCR’s affinity to its cognate allo-HLA and that antigen-antibody interactions are largely governed by electrostatic forces [60]. However, these differences are correlated with the development and level of dnDSAs, similar to eplets [40,42]. More recently, a computational approach that quantifies the surface electrostatic potential difference (or ESM-3D) of allo-HLAs has been shown to improve the prediction of humoral alloimmunity [60].

## 3. T Cell Epitopes: The Second Activating Signal

Immediately following allo-HLA uptake by the recipient’s B cells, allo-HLAs are processed and degraded into antigenic peptides in the endosomal/lysosomal network, then loaded onto the recipient’s HLA class II (HLA-II) molecules and presented to allo-T_fh_ [32]. The processing of allo-HLA molecules generates a pool of allo-HLA-derived peptides (allo-HLApeps) comprised of self and non-self-peptides, which may be loaded onto the recipients’ HLA-II molecules provided their affinity is higher than that of endogenous peptides present within the endosomal/lysosomal network’s peptide pool. The recipient’s HLA-II molecules loaded with allo-HLApep may be exported to the cell surface for the presentation to allo-T_fh_. The recognition of “non-self” allo-HLApeps by the allospecific T cell receptor (allo-TCR) of allo-T_fh_ constitutes the second activation signal mandatory for allospecific naïve B cells to proliferate and differentiate into allospecific IgG-secreting plasma cell and memory B cells [61]. Indeed, allo-T_fh_ induce the production of high-affinity antibodies through the secretion of cytokines (i.e., IL-4 and IL-21) and cell-to-cell interactions [62]. Although mapping the events of allo-HLA processing and HLA-II loading in the endosomal/lysosomal network is not yet fully elucidated, the second activation signal is required to mount a robust humoral allo-immune response [32]; therefore, allo-HLApeps, or HLA T cell epitopes (TCEs), are crucial for the assessment of allo-HLA immunogenicity.

Any recipient’s HLA-II T cell epitope repertoire exhibits a tremendous peptide diversity, particularly in length [63]. Indeed, the HLA-II peptide-binding cleft—opened at both ends—can accommodate peptides of varying length, usually 12 to 25 residues, and sometimes longer peptides or even whole molecules [64]. T cell epitopes are anchored in the HLA-II peptide-binding cleft by a core-binding motif (9 residues long), which is the critical motif impacting TCR and HLA-II/peptide interactions [65]. As a result of thymic selection, the TCR of allo-T_fh_ may recognize “non-self” (or mismatched) core-binding motifs with high affinity, whereas “self” ones are recognized with lower affinity [66]. Only high-affinity binding between the recipient’s HLA-II/TCE complexes and allo-TCRs may activate allo-T_fh_ to secrete different cytokines that promote further activation, differentiation, and proliferation of allospecific naïve B cells. Currently, most mismatched TCEs described represent hypothetical peptides (≥9 mers, Figure 1) that were computationally predicted to bind to the recipients’ HLA-II antigens using the NetMHCIIpan algorithm [67]. Using this algorithm, E. Spierings developed a tool that evaluates the number of HLA class II-restricted predicted indirectly recognizable HLA epitopes (PIRCHE-II) of HLA mismatches [68]. PIRCHE-II corresponds to the number of allo-HLA-derived TCEs with a mismatched nonamer core-binding motif predicted to be presented by the recipient’s HLA-DR molecules. Similar to eplets, the quantitative measure of mismatched TCEs, or PIRCHE-II score, was associated with the development of dnDSAs and renal allograft failure [69]; however, the significant overlap in the PIRCHE-II scores between immunogenic and non-immunogenic allo-HLA indicates that the number of mismatched TCEs alone cannot clearly predict allo-HLA immunogenicity. Indeed, the nature of the allo-HLApeps, “self” or “non-self”, was associated with the development of HLA-II dnDSA renal transplant recipients who experienced an antibody-mediated rejection episode [41].

The quantitative measure of mismatched TCEs has been used to predict dnDSA development and transplant outcomes. Indeed, the PIRCHE-II score strongly predicted dnDSA development, independently from the antigen mismatch [9]. Also, the PIRCHE-II score was associated with an increased risk of T cell-mediated rejection and allograft failure [69,70]. Lastly, the PIRCHE-II is an independent predictor of histopathological progression of immune-mediated injury [71]. Therefore, using TCEs may significantly contribute to identifying acceptable mismatches with decreased risk of dnDSAs and allograft failure and thus improve immunosuppression management and long-term allograft outcomes.

## 4. Limitations

The current strategies to assess and predict allo-HLA immunogenicity remain imprecise and have limitations. Most importantly, HLA immunogenicity assessment relies on HLA antibody detection methods, which have many limitations regarding the accurate detection of pathogenic HLA antibodies, the determination of eplet-specific dnDSAs, and center-to-center variability [72]. Also, current HLA antibody detection methods can detect DSAs against native and non-native, so-called “denatured”, HLA molecules [73], but their respective pathogenic potential remains to be fully elucidated. Until the accuracy of pathogenic dnDSA detection is improved, allo-HLA immunogenicity prediction can only be as precise as the power of HLA antibody detection methods. Although comparative studies evaluating the accuracy of HLA antibody detection methods from different manufacturers could improve the detection of pathogenic DSAs [74], large-scale clinical studies need to be performed to confirm the precision of pathogenic DSA identification. Secondly, although the number of mismatched eplets is associated with dnDSA development, roughly 80% of recipients with a high mismatched eplet load do not develop dnDSAs, and the absence of mismatched, corresponding to 14% of the patients, absence of eplet mismatches was associated with 100% dnDSA-free survival [46]. Therefore, dnDSA development predictions, based on mismatched eplets, accurately determine permissible mismatches but not immunogenic ones. In a retrospective study of 236 kidney recipients transplanted, the development of dnDSAs could be predicted accurately, but mostly for allo-HLA that may not induce a dnDSA [37]. Because the presence of a mismatched eplets is the first biological prerequisite for dnDSA development, assessment of allo-HLA immunogenicity should be reevaluated, excluding allo-HLAs without mismatched eplets to truly characterize and predict mismatches that will induce dnDSAs instead of the ones that do not. Thirdly, allo-HLA immunogenicity assessed with TCEs is limited by its inherent computational approach [75,76]. Although databases listing peptide sequences that are known to be presented by HLA-II antigens could help validate computationally predicted TCEs, the complexity of HLA-II presentation pathways remains a major hurdle [41]. Indeed, several parameters that contribute to the characterization of computationally defined TCEs have not been evaluated for their impact on allo-HLA immunogenicity assessment, which include: HLA-DR, HLA-DQA1/DQB1, and HLA-DPA1/DPB1 peptide presentation; peptide length (from 9 mers to whole molecules); peptide-binding affinity measurements (IC_50_ vs. percentile ranks) and thresholds; and the presence of endogenous peptides. Recent findings have predicted allo-HLA immunogenicity using both B and T cell allorecognition, yet the study population was not an organ transplant recipient population that did not receive immunosuppression treatment and with at most one haplotype mismatch [42]. Despite an improved prediction of allo-HLA immunogenicity, extrapolating the conclusion to a transplant setting should be taken with caution, mainly because of the impact of immunosuppression on dnDSA development [13]. Lastly, several other factors, including under-immunosuppression, non-adherence to treatment, or allograft inflammation (i.e., TCMR) [6,9,10], can increase HLA immunogenicity and promote the development of dnDSAs. Also, the lack of specific quantification of HLA expression levels from an allele-specific basis, which can vary from one individual to another, can also significantly impact the immunogenicity of allo-HLAs. These factors are rarely considered when predicting the development of dnDSAs, although this immunological phenomenon is progressive and affected by extrinsic factors other than HLA mismatches.

## 5. Future Directions

As described in Figure 2, several major aspects of the indirect allorecognition pathway have yet to be examined for their impact on allo-HLA immunogenicity assessment. With regards to eplets, only 146/521 (28.0%) of eplets are antibody-verified (HLA-ABC: 72/252 (28.6%); HLA-DRB: 36/124 (29.0%); HLA-DQ: 27/83 (32.5%); and HLA-DP: 11/62 (17.7%)) (as of April 2024, www.epregistry.com.br (accessed on 2 April 2024)), which warrants further experimental validation to confirm whether eplets are immunogenic and potentially pathogenic in transplant recipients. Indeed, not all eplets are exhibited by native heterotrimeric HLA molecules; they also present other HLA isoforms (HLA monomer and heterodimer) [48,73]. Thus, the biological relevancy of all eplets remains to be fully elucidated, particularly because different soluble HLA (sHLA) isoforms may expose/create eplets that are immunogenic even in non-alloimmunized individuals [77]. Transplant recipients may develop dnDSAs directed against non-heterotrimeric HLA isoforms, which do not seem deleterious for allograft survival [73,78,79]. Hence, such dnDSAs may impair the ability to predict allo-HLA immunogenicity, particularly for predicting pathogenic dnDSAs development. Indeed, the implementation of eplets in the clinics is dependent on the verification of DSA at an eplet level. In this regard, novel recombinant human HLA-specific mAbs were generated to confirm two HLA-I eplets and seven HLA-II eplets, including the verification of three previously non-verified eplets [53].

Furthermore, it is paramount to refine the HLA immunogenicity further, as the prediction of pathogenic dnDSAs, with an impact on long-term allograft outcomes, remains imprecise. Since non-alloimmunized individuals can produce DSA and auto-HLA antibodies [77], there are clearly anti-HLA antibodies detected by current state-of-the-art methods that do not seem to have any pathological impact. Consequently, there is an imminent need for the refinement of HLA antibody detection, particularly to characterize dnDSAs in terms of their eplets rather than antigenic specificity. The recent development of novel HLA antibody screening reagents allows targeting individual HLA antigens to explore and define DSA reactivity at the antigen-specific level. Lastly, evaluating the physiochemical difference of mismatched eplets, instead of amino acids or whole HLA antigens, may characterize eplets with a strong antigenic potential since allo-BCR/eplet interactions are largely governed by electrostatic forces [60].

Regarding TCEs, the processing of allo-HLAs in the endosomal/lysosomal compartments of allospecific B cells may influence allo-HLA immunogenicity, yet very little is known [41]. This process shapes the allo-HLA TCE repertoire, and TCEs can be destroyed if they contain the cleavage site of any proteases [80]. Paradoxically, HLA molecules are inherently resistant to the action of different proteases. Therefore, elucidating the cleavage pattern of allo-HLAs by endosomal/lysosomal proteases may further refine the prediction of dnDSAs using TCEs. Similar to eplets, sHLA isoforms may change the TCE repertoire, consequently impacting humoral alloimmunity. For HLA class I (HLA-I) antigens, TCEs may be found on the HLA heavy chain or the β2microglobulin; for HLA-II antigens, TCEs may be found on both α and β chains. If a TCE, derived from a self-component (β2microglbulin or DRα1), is preferentially loaded on the recipients’ HLA-II antigens and presented to allo-T_fh_, allospecific B cells should not receive the second activating signal required for dnDSA development. Interestingly, HLA-DQ antigens are more immunogenic than HLA-I and HLA-DR antigens, possibly because HLA-DQ antigens are composed of polymorphic α and β chains [15,81]. In addition, evaluating the physiochemical difference of mismatched TCEs may characterize TCEs with a strong immunogenic potential. Last, the impact of the recipient’s HLA-II phenotype on dnDSA development is not well studied [41], although a recipient’s HLA-II TCE repertoire may confer a resistance or a susceptibility for developing a dnDSA against a specific allo-HLA by binding preferentially “self” TCEs or “non-self” TCEs, respectively. In addition, several aspects of the recipient’s HLA-II presentation, such as HLA-DR vs. -DQ vs. -DP TCE presentation, TCE peptide lengths, and TCE-binding affinity thresholds, remain to be thoroughly investigated for their impact on allo-HLA immunogenicity. Artificial intelligence (AI) is a promising technology that can be used to predict the development of dnDSAs in transplant patients to enhance pre-transplant risk assessment and post-transplant monitoring. AI could identify complex patterns and predictors of DSA development. However, the use of AI to predict allo-HLA immunogenicity remains limited.

The field of HLA immunogenicity will significantly benefit from developing novel in vitro models that will enhance pre-transplantation immunological risk assessment and post-transplantation monitoring. Conventional in vitro models to study the alloimmune response employ 2D platforms (e.g., mixed leukocyte reaction) and animal models (e.g., skin rejection models). However, these methods present significant challenges due to their poor physiological relevance and clinical translation [82,83]. Two-dimensional culture systems do not provide the cell-extracellular matrix (ECM) and cell-cell interactions that regulate the functional phenotype of cells in vivo. Similarly, animal models suffer from significant histocompatibility and immunogenetic incompatibilities with humans. Novel models incorporating advanced biotechnological tools such as organoids, microfluidic systems, and biofabricated scaffolds to mimic the complex cellular interactions and physiological functions of human organs have been developed. With the recent passing of the FDA Modernization Act 2.0, organ-on-a-chip (OoC) technologies have gained tremendous attraction for their capability to accurately represent the human body and generate clinically translatable data [84,85,86,87,88,89]. Several OoC models of human organs have been manufactured on chips [89,90,91,92]; however, no existing model allows for studying allograft rejection or HLA immunogenicity despite the different OoC models available. For example, modeling antibody-mediated rejection on-a-chip requires a microfluidic device that mimics the physiological function of an organ’s microvasculature, since the donor’s endothelium, a crucial determinant in allograft rejection mechanisms, is the first contact site with the recipient’s immune system [93,94,95,96]. Recent OoC models included a microvasculature to mimic the tissue microenvironment better and study endothelial function and simple T cell migration across endothelial monolayer [97,98]; however, no vascularized OoC platforms have been established to model allograft rejection. Lastly, OoC platforms are easily amenable to using patient-derived cells to offer a solution to tailor immunosuppression treatment strategies to individual recipients for better long-term outcomes. Leveraging these in vitro technologies to advance the understanding of HLA immunogenicity will be paramount to elucidating the immunogenic potential of allo-HLA in a personalized manner.

## 6. Conclusions

The immunogenicity of allo-HLAs is a simple concept disguising a complex process, which involves several stages defined as a succession of events that may drive or prevent the activation, differentiation, and proliferation of B cells, eventually leading to the development of IgG dnDSAs (Figure 3). Several aspects of the biological process that dictate allo-HLA immunogenicity have been extensively investigated, yet the assessment of allo-HLA immunogenicity and the prediction of dnDSAs is not fully elucidated and remains imprecise. Despite all the efforts to predict humoral alloimmunity, a comprehensive assessment of allo-HLA immunogenicity is still warranted. Indeed, the prediction of humoral alloimmunity seems to efficiently predict who is not at risk of developing a dnDSA rather than predicting who is at risk, and this distinction has a major impact in promoting permissible HLA mismatched organ transplantations, organ allocation scheme, personalized immunosuppression, and clinical trial design. It must be realized that the assessment of allo-HLA immunogenicity only characterizes allo-HLAs that will not induce dnDSAs. A comprehensive analysis of all the parameters that can be used to characterize allo-HLAs, both molecular differences and recipient’s characteristics (HLA-II phenotype and immunosuppressive regiments), may improve the characterization of allo-HLAs that will induce dnDSAs.

## Figures and Tables

**Figure 1 antibodies-13-00061-f001:**
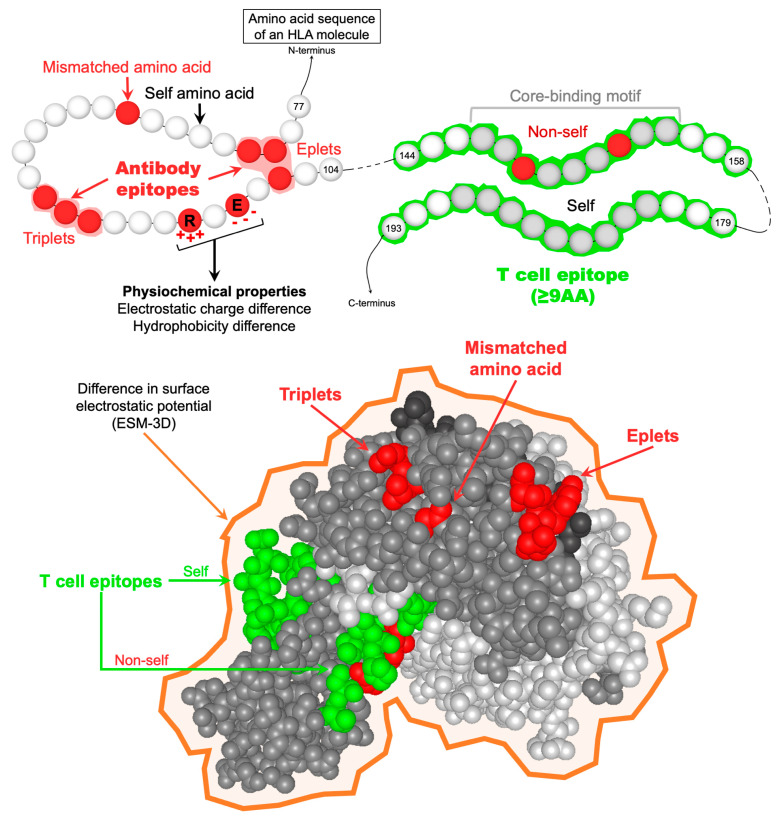
Representation of the different parameters used to describe allo-HLAs from a molecular perspective. HLA antibody epitopes are defined by at least one mismatched amino acid in a continuous (triplet) or discontinuous (eplet) configuration and exposed or not on the molecular surface. From an allo-BCR perspective, allo-HLAs are also described by their physiochemical properties, namely the number of mismatched amino acids, the electrostatic and hydrophobicity charge difference, and the difference in surface electrostatic potential (ESM-3D). TCEs are defined as the predicted allo-HLApep that binds to the recipients’ HLA-II antigens. TCEs should be at least 9 amino acids long, corresponding to the HLA-II core-binding motif responsible for anchoring peptides in the HLA-II peptide-binding cleft. TCEs are considered “non-self” only when the core-binding motif carries at least one mismatched amino acid residue. *Abbreviations:* AA: amino acid; allo-BCR: allospecific B cell receptor; allo-HLAs: allogeneic HLA antigens; allo-HLApeps: allo-HLA-derived peptides; HLA-II: HLA class II; TCEs: HLA T cell epitopes.

**Figure 2 antibodies-13-00061-f002:**
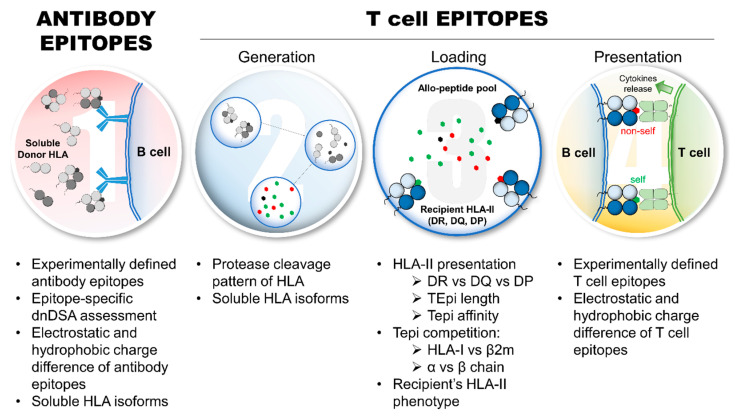
Aspects of allo-HLA immunogenicity that require further investigations to refine and improve the assessment of HLA immunogenicity and predict pathogenic dnDSA development. Eplet requires further experimental validations and investigations regarding immunogenicity and pathogenicity in transplant recipients, which includes: the refinement of HLA antibody detection, particularly epitope-specific dnDSAs; the interference of dnDSAs directed against non-heterotrimeric sHLA isoforms; and the relevance of electrostatic and hydrophobic charge differences of eplets on allo-HLA immunogenicity. From a TCE perspective, several aspects deserve further investigations and experimental validations, particularly the protease cleavage pattern of allo-HLA and the generation of TCEs, the influence of sHLA isoforms on TCE repertoire and TCE competition (HLA-I: heavy chain vs. β2; HLA-II: α vs. β chains), the recipient’s HLA-II presentation parameters, such as HLA-DR vs. -DQ vs. -DP presentation, TCE peptide length, and binding affinity thresholds, and the relevance of electrostatic and hydrophobic charge differences of experimentally validated TCEs on allo-HLA immunogenicity. *Abbreviations:* allo-HLAs: allogeneic HLA antigens; β2m: β2microglobulin; dnDSAs: de novo donor-specific HLA antibodies; HLA-I: HLA class I; HLA-II: HLA class II; sHLA: soluble HLA; TCE: HLA T cell epitope.

**Figure 3 antibodies-13-00061-f003:**
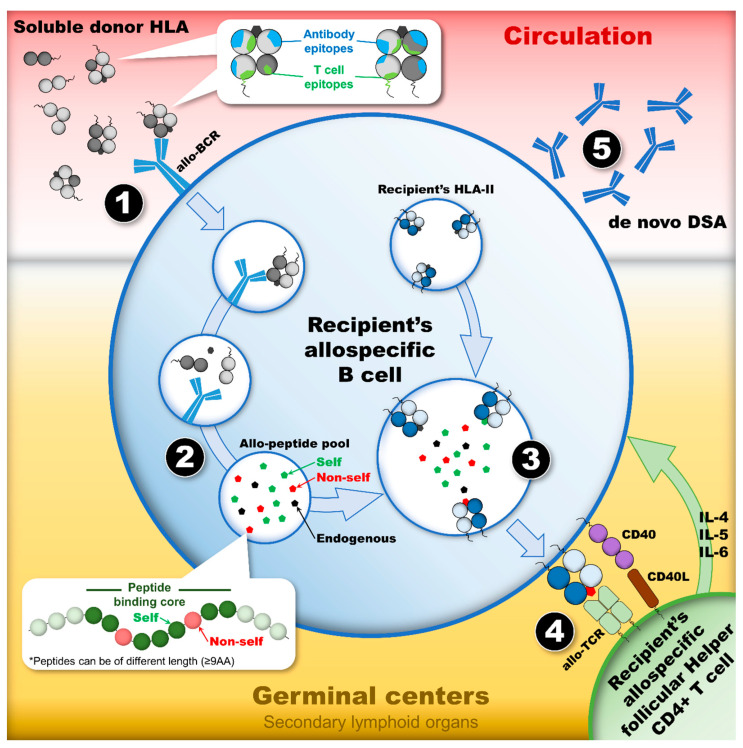
The humoral alloimmune response, or the indirect allorecognition pathway, leads to the development of dnDSAs provided there are appropriate activating signals. This dynamic process is a cascade of events in which four stages constitute the checkpoints of the humoral response activation. (1) The first stage occurs in circulation where a soluble allo-HLA is recognized by the allo-BCR of a recipient’s naïve B cell. This recognition, which initiates the humoral response by internalizing a soluble allo-HLA, relies on the presence of mismatched eplets expressed on the surface of allo-HLA. (2) After the internalization of allo-HLAs, the second stage constitutes the processing and degradation of allo-HLAs in the endosomal/lysosomal compartments of the recipient’s allospecific B cell. The proteases’ cleavage pattern of allo-HLAs into small antigenic peptides generates the TCE pool. (3) The third stage constitutes the loading of TCEs onto the recipient’s HLA-II antigens, which are exposed to a T cell epitope pool containing “non-self” and “self” TCEs, as well as endogenous peptides. Only the ones with the highest affinity for the recipients’ HLA-II peptide-binding cleft will be loaded preferentially. (4) After the loading of TCEs on the recipient’s HLA-II antigens and their export to the cell surface, the fourth stage constitutes the presentation of TCEs to allo-T_fh_. The nature of the TCEs presented dictates whether allo-T_fh_ can provide the necessary co-stimulatory signal to induce allospecific B cell proliferation and differentiation into allospecific memory B cell and IgG-secreting plasma cells. The presentation of “non-self” TCEs may lead to the formation of dnDSAs, whereas the presentation of “self” TCEs or endogenous T cell epitopes may not. *Abbreviations:* allo-BCR: allospecific B cell receptor; allo-HLAs: allogeneic HLA antigens; allo-Tfhs: allo-HLA-specific follicular helper CD4+ T cells; dnDSAs: de novo donor-specific HLA antibodies; HLA-II: HLA class II; TCE: HLA T cell epitope.

## Data Availability

Not applicable.

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
