# Peer review of "Allogeneic HLA Humoral Immunogenicity and the Prediction of Donor-Specific HLA Antibody Development"

_2073-4468, 2024, doi:10.3390/antib13030061_

Round 1

Reviewer 1 Report

Comments and Suggestions for Authors

This review paper is fully discussed the advantages of the allogenic HLA humoral immunogenicity, its original from, its application, the recent prediction algorithms, and the limitations. The cartoons illustrate the circulation process very clearly and helpful for the readers to understand straightforward, initiating from soluble donor HLA interaction with allo-BCR, then going through the endocytosis or lysocytosis in the recipient’s allo-specific B cells, resulting in a generation of TEpi (HLA T cell epitope) pool. The recipients’ MHC class II interacts with TEpi peptide, which will be presented to the allo-HLA-specific follicular helper CD4+ T cells, resulting in the dnDSA (de novo donor-specific HLA antibodies) formation in the presence of specific TEpi peptide recognition.

This review paper lets colleagues know the current research progress of allo-HLA immunogenicity and points out the fact that the algorithm can predict which one will not be at risk. This will contribute to allowable HLA-mismatched organ transplantations and clinical trial design.

I am wondering if it is possible that

(1) add the negatively charged residues and positively charged residues symbols in the figure 1 since they talked about the electrostatic and hydrophobicity charge differences. 

(2) Tfh cells are special subsets of  CD4+ T cells that are identified in human tonsil. Can this be applied to other conventional CD4+ T cells as well? 

Author Response

Comment 1: Add the negatively charged residues and positively charged residues symbols in the figure 1 since they talked about the electrostatic and hydrophobicity charge differences.

Response 1: Thank you for pointing this out. I agree with the comment. Therefore, I have modified Figure 1 (Page 4, Line 174) accordingly to represent the negatively and positively charged residues that are related to electrostatic and hydrophobicity charge differences.

Comment 2: Tfh cells are special subsets of CD4+ T cells that are identified in human tonsil. Can this be applied to other conventional CD4+ T cells as well?

Response 2: Thank you for the comment. Although follicular helper CD4+ T cells (Tfh) were first identified in human tonsils, they are generally located in secondary lymphoid organs, such as the spleen and lymph nodes (https://doi.org/10.1038/s41467-023-39299-3). Therefore, this cannot be applied to other conventional CD4+ T cells. In this review, we focus on Tfh located in the germinal centers within the B cell zone of secondary lymphoid organs. Accordingly, I have revised the Introduction as follows: “This humoral immune response is dependent on the ability of B cells to capture and process allo-HLA into antigenic peptides that are subsequently presented to follicular helper CD4+ T cells (Tfh), located in secondary lymphoid organs, such as the spleen, tonsils, and lymph nodes. The cognate CD4+ T cell helper function is required to form germinal centers (GCs) where B cells differentiate into IgG dnDSA-secreting plasma cells and memory B cells [32,33].” (Page 2, Line 54-59).

Reviewer 2 Report

Comments and Suggestions for Authors

The review entitled "Towards understanding allogeneic HLA humoral immunogenicity: a simple concept disguising a complex process" summarizes new views on the development of donor HLA specific antibodies with a special focus on computational strategies.

In this detailed review the author nicely presents current insights in the immunogenicity of allogenic HLA in the transplantation setting. Several aspects of the biological process of the development of de novo donor specific HLA antibodies are described in detail. The content is of high scientific interest for the community and represents new results and approaches.

The author should consider a more precise title for his review. 

Author Response

Comment 1: The author should consider a more precise title for his review.

Response 1: Thank you for the comment. I have slightly modified the title to reflect more precisely the content of this review. The new title is “Allogenic HLA humoral immunogenicity and the prediction of donor-specific HLA antibody development” (Page 1, Line 2-3).

Reviewer 3 Report

Comments and Suggestions for Authors

In this review entitled, “Towards understanding allogeneic HLA humoral immunogenicity: a simple concept disguising a complex process” by Vadim Jucaud, highlights the importance of de novo donor-specific HLA antibodies (dnDSA) following solid organ transplantation and signifies importance of the tools to predict the dnDSA development. Overall, despite the importance of the topic, the review fails to provide a cohesive overview of this topic and lacks a clear organization which hampers the flow of the information.

Major Comments;

1. The introduction section of the review, should be updated to add new section (e.g. Germinal center biology and antibody secretion providing set up for what is later going to be presented on later on T cell epitopes, antibody epitopes.

2. I suggest adding  a section on the pre-translational in-vitro models (organ-on-a-chip) to assess allogeneic HLA humoral immunogenicity.

Minor comments:

1. Something is missing between the sentence in lines 45-49 and the one in lines 49-51. Further context in the utility of computational tools should be provided.

2. Lines 45-49: The references are provided in two brackets next to each other at the end of this sentence. Any specific reasons to separate the two?

Comments on the Quality of English Language

Editing to improve readability and cohesiveness

Author Response

Comment 1: The introduction section of the review should be updated to add new section (e.g. Germinal center biology and antibody secretion providing set up for what is later going to be presented on later on T cell epitopes, antibody epitopes.

Response 1: I thank the reviewer for this comment. I agree with the comment; therefore, I added a new section to the introduction describing the germinal center biology and their role in the production of donor-specific antibodies. The introduction was revised as follows: “GCs are specialized microenvironments within secondary lymphoid organs where mature B cells undergo proliferation, somatic hypermutation, and selection for high-affinity antibodies [32,33]. In GCs, the generation of dnDSA is orchestrated through a complex interplay of cellular and molecular mechanisms. The process begins when allo-HLA are captured and internalized by naive B cells. Upon recognizing allo-HLA via their B cell receptors, B cells internalize, process, and present allo-HLA-derived peptides on their surface in the context of HLA class II molecules. This antigen presentation is crucial for the subsequent interaction with Tfh, which provides essential signals for B cell activation and differentiation. B cells proliferate rapidly within the GC and undergo somatic hypermutation of their immunoglobulin genes, mediated by the enzyme acti-vation-induced cytidine deaminase [34,35]. This process generates a diverse repertoire of B cells with varying affinities for the allo-HLA. Tfh cells provide critical survival and differentiation signals through the engagement of CD40 ligand (CD40L) on Tfh cells with CD40 on B cells and the secretion of cytokines (i.e., IL-21 and IL-4) [36]. These interactions between Tfh and B cells promote the selection of high-affinity B cells and their differen-tiation into long-lived plasma cells, which secrete dnDSA, or memory B cells, which provide a rapid and robust response upon re-exposure to allo-HLA.“ (Page 2, Line 60-76).

Comment 2: I suggest adding a section on the pre-translational in-vitro models (organ-on-a-chip) to assess allogeneic HLA humoral immunogenicity.

Response 2: I thank the reviewer for this comment. I revised the last paragraph of the Future Directions section to describe in more details how the current pre-clinical models to assess allogeneic HLA humoral immunogenicity are imprecise, leading to the description of novel in vitro models (i.e., Organ on-a-chips) that could be used to assess allogeneic HLA humoral immunogenicity more precisely in the future. The revised paragraph is as follows: “Conventional in vitro models to study the alloimmune response employ 2D platforms (e.g., mixed leukocyte reaction) and animal models (e.g., skin rejection models). How-ever, these methods present significant challenges due to their poor physiological rele-vance and clinical translation [83,84]. 2D culture systems do not provide the cell-extracellular matrix (ECM) and cell-cell interactions that regulate the functional phenotype of cells in vivo. Similarly, animal models suffer from significant histocom-patibility and immunogenetic incompatibilities with humans.“ (Page 9, Line 384-390).

Comment 3: Something is missing between the sentence in lines 45-49 and the one in lines 49-51. Further context in the utility of computational tools should be provided.

Response 3: I thank the reviewer for this comment. I modified the sentence starting in line 49 to improve the connection of the two statements regarding the utility of computational tools to predict the development of DSA. The modified sentence is as follows: “Recognizing the complexity and variability of donor-specific humoral responses, several computational tools were developed, integrating large amounts of immunogenetic and molecular data, to assess the immunogenicity of allo-HLA and to identify permissible mismatches that do not induce dnDSA [26,27].“ (Page 2, Line 49-52).

Comment 4: Lines 45-49: The references are provided in two brackets next to each other at the end of this sentence. Any specific reasons to separate the two?

Response 4: Thank you for pointing this out. There were no specific reasons to separate the two references, it was a typo. Therefore, I combined the references as follows: “[6,10,21-25]” (Page 2, Line 49).

Reviewer 4 Report

Comments and Suggestions for Authors

This is a nice review on humoral alloimmunity after solid organ transplantation, describing the known mechanisms of allorecognition from a B- and T-cell perspective. The topic is ver interesting and future perspectives include refinements and improvements of current predictive tools.

I have few suggestions:

- “AbEpi” is quite unusual and the term “eplet” already exists to define what the Author here calls AbEpi: for this reason the term “AbEpi” should be replaced by “eplet”;

- the same is true for “TEpi”, to be replaced by the more common “TCE”;

- line 327: “an pathological impact” maybe is “any” or “a” ? Please correct;

- the chapter “Future directions” should be shortened and be more focused on what Author thinks the future directions are: specifically I suggest to include some statements about Artificial Intelligence in this setting and also to discuss about the potential advantages of increasing the 14% rate of eplet-matched transplants (ref. 44) by implementing allocation algorithms. This would improve the rate of dnDSA-free survival;

- although not the focus of this review, the NK-mediated rejection (i.e. Koenig A et al., Nature Communication 2019) has to be at least cited, perhaps in the Introduction to put humoral immunogenicity in the context of the wider field of allorecognition.

Author Response

Comment 1: “AbEpi” is quite unusual and the term “eplet” already exists to define what the Author here calls AbEpi: for this reason the term “AbEpi” should be replaced by “eplet”;

Response 1: I thank the reviewer for this comment. I replaced “AbEpi” by “eplet” throughout the manuscript.

Comment 2: the same is true for “TEpi”, to be replaced by the more common “TCE”;

Response 2: I thank the reviewer for this comment. I replaced “TEpi” by the more common “TCE” throughout the manuscript.

Comment 3: line 327: “an pathological impact” maybe is “any” or “a” ? Please correct;

Response 3: Thank you for pointing this out. I revised the manuscript as follows: “any” (Page 8, line 347).

Comment 4: the chapter “Future directions” should be shortened and be more focused on what Author thinks the future directions are: specifically I suggest to include some statements about Artificial Intelligence in this setting and also to discuss about the potential advantages of increasing the 14% rate of eplet-matched transplants (ref. 44) by implementing allocation algorithms. This would improve the rate of dnDSA-free survival;

Response 4: Thank you for the comment. I revised the third paragraph of the Future Direction section to include some statement about artificial intelligence. The revised paragraph is as follows: “Artificial intelligence (AI) is a promising technology that can be used to predict the development of dnDSAs in transplant patients to enhance pre-transplant risk assessment and post-transplant monitoring. AI could identify complex patterns and predictors of DSA development. However, the use of AI to predict allo-HLA immunogenicity remains limited.“ (Page 9, Line 377-381).

Comment 5: although not the focus of this review, the NK-mediated rejection (i.e. Koenig A et al., Nature Communication 2019) has to be at least cited, perhaps in the Introduction to put humoral immunogenicity in the context of the wider field of allorecognition.

Response 5: Thank you for the comment. Although I recognize the importance of the wider field of allorecognition and the importance of NK cell-mediated rejection, as elegantly described by Koenig A et al., Nature Communication 2019, I feel that incorporating other aspect of allorecognition would outside of the scope of this review, which focuses only on humoral rejection.

Round 2

Reviewer 3 Report

Comments and Suggestions for Authors

Thank you for the revisions made.

Comments on the Quality of English Language

Minor edit